# Common coupling map advances GPCR-G protein selectivity

Alexander S Hauser[1], Charlotte Avet[2], Claire Normand[3], Arturo Mancini[3], Asuka Inoue[4], Michel Bouvier[2]*, David E Gloriam[1]*

[1]Department of Drug Design and Pharmacology, University of Copenhagen, Copenhagen, Denmark; [2]Institute for Research in Immunology and Cancer (IRIC), and Department of Biochemistry and Molecular Medicine, Université de Montréal, Montréal, Canada; [3]Domain Therapeutics North America, Montréal, Canada; [4]Graduate School of Pharmaceutical Sciences, Tohoku University, Sendai, Japan

**Abstract** Two-thirds of human hormones and one-third of clinical drugs act on membrane receptors that couple to G proteins to achieve appropriate functional responses. While G protein transducers from literature are annotated in the Guide to Pharmacology database, two recent large-scale datasets now expand the receptor-G protein 'couplome'. However, these three datasets differ in scope and reported G protein couplings giving different coverage and conclusions on G protein-coupled receptor (GPCR)-G protein signaling. Here, we report a common coupling map uncovering novel couplings supported by both large-scale studies, the selectivity/promiscuity of GPCRs and G proteins, and how the co-coupling and co-expression of G proteins compare to the families from phylogenetic relationships. The coupling map and insights on GPCR-G protein selectivity will catalyze advances in receptor research and cellular signaling toward the exploitation of G protein signaling bias in design of safer drugs.

*For correspondence:
michel.bouvier@umontreal.ca (MB);
david.gloriam@sund.ku.dk (DEG)

## Editor's evaluation

Using data sets are from three distinct sources, the authors present a meta-analysis to arrive at a consensus for G protein-coupled receptor (GPCR) coupling specificity to G proteins, as well as to identify and highlight differences or incongruencies. Compiling these data sets into a unified format will be extremely useful for investigators to understand receptor and effector relationships. The meta-analysis will help to deconvolute the complex physiology and pharmacology underlying hormone or drug actions acting on receptor superfamilies, and a better understanding of receptor-G protein promiscuity will ultimately help in identifying safer therapeutics.

## Introduction

G protein-coupled receptors (GPCRs) represent the largest family of proteins involved in signal propagation across biological membranes. They recognize a vast diversity of signals going from photons and odors to neurotransmitters, hormones, and cytokines (*Armstrong et al., 2020*). Their main signaling modality involves the engagement and activation of G proteins. G proteins are heterotrimeric proteins consisting of α, β, and γ subunits that dissociate to α and βγ upon activation by a GPCR. G proteins are named by their α subunit (16 in human) and are divided into four families which share homology and downstream signaling pathways: $G_s$ ($G_s$ and $G_{olf}$), $G_{i/o}$ ($G_{i1}$, $G_{i2}$, $G_{i3}$, $G_o$, $G_z$, $G_{t1}$, $G_{t2}$' and $G_{gust}$), $G_{q/11}$ ($G_q$, $G_{11}$, $G_{14}$, and $G_{15}$), and $G_{12/13}$ ($G_{12}$ and $G_{13}$). A GPCR's activation of G proteins can be very selective or promiscuous and change upon ligand-dependent biased signaling that alters its profile on the G

protein subtype or family levels. The pleiotropic signaling and ligand-dependent bias of GPCRs pose a grand challenge in human biology to map the differential activation of specific G proteins.

The International Union of Basic and Clinical Pharmacology (IUPHAR)/British Pharmacological Society (BPS) Guide to Pharmacology (GtP) database contains reference data from expert curation of literature (*Armstrong et al., 2020*). GtP couplings cover 253 GPCRs and the 4 G protein families. The G protein families have been classified into 'primary' and 'secondary' transducers without quantitative values. Recently, the Inoue group determined the first large-scale quantitative coupling profiles of 148 GPCRs to the $G_q$ wildtype and 10 G protein chimeras employing a transforming growth factor-α (TGF-α) shedding assay (neurotensin 1 [$NTS_1$] and thyrotropin-releasing hormone [$TRH_1$] added herein making it 150 receptors) (*Inoue et al., 2019*). Those chimeras consist of a $G_q$ backbone in which the six most C-terminal Gα residues – a part of the H5 domain inserting to the intracellular receptor cavity – have been replaced to represent all 16 human G proteins (five of which have identical sequences to other G proteins, see below). In a paper accompanying the present analysis, the Bouvier group quantified the couplings of 100 GPCRs to 12 G proteins: $G_s$, $G_{i1}$, $G_{i2}$, $G_o$ ($G_{oA}$ and $G_{oB}$ isoforms), $G_z$, $G_q$, $G_{11}$, $G_{14}$, $G_{15}$, $G_{12}$, and $G_{13}$ but not $G_{olf}$ (couples mainly to olfactory receptors), $G_{i3}$, and $G_{t1-2}$ (transducin, couples to rhodopsin [visual] receptors) and $G_{gust}$ (gustducin, couples to taste receptors) (*Avet et al., 2020*). The authors used novel enhanced bystander bioluminescence resonance energy transfer biosensors that allow to monitor G protein activation (G protein Effector Membrane Translocation assay) without need to modify the G protein subunits (except for $G_s$) or the receptors.

Here, we analyze the GtP, Inoue, and Bouvier coupling datasets to determine confident couplings supported by at least two independent sources, including novel couplings discovered jointly by the two latter sources. We establish a scalable protocol to normalize quantitative G protein couplings, combine $E_{max}$ and $EC_{50}$ into a common $\log(E_{max}/EC_{50})$ (*Kenakin, 2017*) value, and aggregate subtypes to allow comparisons across G protein families. On this basis, we develop a unified map of GPCR-G protein couplings that can be filtered or intersected in GproteinDb (*Pándy-Szekeres et al., 2022*), describe GPCR-G protein selectivity across an unprecedented number of receptors and coupling data points, and reveal correlated co-couplings.

## Results

### Tools and resources – common coupling map unifying GPCR-G protein data

#### Systematic profiling doubles the average number of G protein family couplings of GPCRs

To obtain an overview of the coverages of GPCR-G protein coupling sources, we compared all couplings reported by both the Bouvier (*Avet et al., 2020*) and Inoue (*Inoue et al., 2019*) groups and annotated in the GtP database (*Armstrong et al., 2020*; *Figure 1*; *Table 1*). This shows that the three sources together comprise couplings for 265 (67%) out of the 398 nonolfactory GPCRs, and that 70 of these receptors are present in all datasets (*Figure 1B*). The Bouvier and Inoue datasets have collectively quantified individual G protein couplings of 178 receptors using one assay, whereas the remaining 87 receptors have so far only been annotated in GtP on the G protein family level from a multitude of publications and assays.

To allow the comparison of coupling densities and distributions across datasets, we selected the $E_{max}$ threshold (1.4 standard deviations [SDs] above basal) that gives the best agreement between the Bouvier and Inoue dataset. We believe that this is the best possible means to estimate what is correct data (rather than false negative/positive couplings), as large-scale information about what G proteins and GPCRs couple physiologically is not available. This cut-off is more stringent than the minimum of 3% signal above basal used in Inoue's original report (*Inoue et al., 2019*). We also aggregated G protein subtype couplings of the families (see Materials and methods). This reveals that while GtP covers the largest number of receptors they have relatively few couplings – only 38% of all GPCR-G protein family pairs are 'couplers' compared to 65% in the Bouvier and 75% in the Inoue dataset (average of 1.5, 2.6, and 3.0 G protein families per GPCR, respectively; *Figure 1C*). In particular $G_{12/13}$ couplings are underrepresented in GtP where they account for 3% of GPCR-G protein pair datapoints compared to 13% in Bouvier and 17% in Inoue. The Bouvier and Inoue datasets share a more similar overall distribution of couplings, expect for $G_s$ coupling which is twice as frequent within the latter

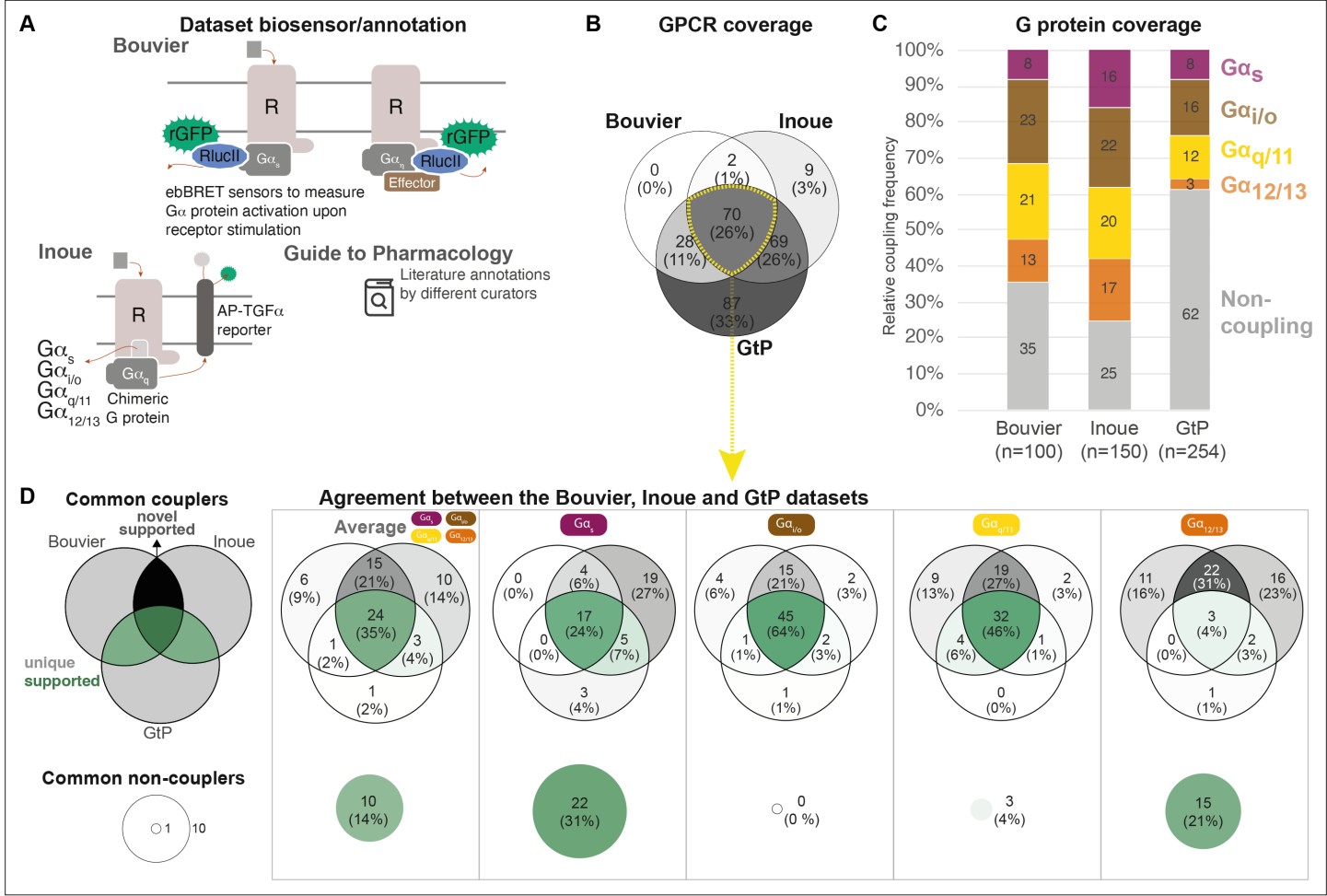

**Figure 1.** Coverage and agreement of the Bouvier lab, Inoue group, and Guide to Pharmacology (GtP) datasets. (**A**) Biosensor principles used by the Bouvier and Inoue groups, and literature annotation stored in the GtP database. (**B**) Intersection of the G protein-coupled receptors (GPCRs) included in the Bouvier (n = 100), Inoue (n = 150), and GtP (n = 254) datasets. (**C**) Relative family distributions of G protein couplings across datasets. (**D**) Comparison of the G protein family coupling profiles of 70 GPCRs present in all of the Bouvier, Inoue, and GtP datasets. More detailed analysis of common G protein couplings for just the Bouvier and Inoue datasets is given in *Figure 1—figure supplement 1*. (**A–D**) Note: All analyses herein cover the 12 G proteins: $G_s$, $G_{i1}$, $G_{i2}$, $G_{oA}$, $G_{oB}$, $G_z$, $G_q$, $G_{11}$, $G_{14}$, $G_{15}$, $G_{12}$, and $G_{13}$. $G_{olf}$ and $G_{i3}$ could not be analyzed, as they had not been tested by the Bouvier group. The Inoue data for the pairs $G_{i1}$-$G_{i2}$, $G_{oA}$-$G_{oB}$, and $G_q$-$G_{11}$ were generated with identical chimera inserting the Gα C-terminal hexamer into a $G_q$ backbone (*Inoue et al., 2019*) meaning that identical datapoints were used to assess their coverage and agreement with other datasets (*Table 1*).

The online version of this article includes the following figure supplement(s) for figure 1:

**Figure supplement 1.** Common G protein-coupled receptor (GPCR-G) protein couplings in the Bouvier and Inoue datasets.

dataset (16% compared to 8%). This demonstrates that the two first systematic coupling profiling studies have substantially expanded the known GPCR-G protein 'couplome' and that their assay platforms are amenable to high-throughput profiling also for the $G_{12/13}$ family for which robust activation assays appeared only recently (*Quoyer et al., 2013*; *Olsen et al., 2020*; *Maziarz et al., 2020*).

Given that the Bouvier and Inoue groups used different biosensors and GtP annotates literature reports from very diverse assays, we sought to determine to which extent they report the same couplings for the same GPCRs, that is, their 70 common receptors (data in tab 'BIG-QualComp' in *Source data 3*). We find that all three sources agree on an average of 49% of G protein family couplings/noncouplings (distributed as $G_s$: 55%, $G_{i/o}$: 64%, $G_{q/11}$: 50%, and $G_{12/13}$: 25%, *Figure 1D*). When instead analyzing the quantitative studies alone (excluding GtP), the agreement increases to 70% across G protein families. This agreement is 68% for individual G protein datapoints, which display an even larger span ranging from at least 53% for $G_{15}$ up to 81% for $G_{oA}$ (*Figure 1—figure*

**Table 1.** G proteins tested by Bouvier et al. and Inoue et al.
The two published datasets contain 11 common G proteins (G$_s$, G$_{i1}$, G$_{i2}$, G$_o$, G$_z$, G$_q$, G$_{11}$, G$_{14}$, G$_{15}$, G$_{12}$, and G$_{13}$), two Inoue et al. specific G proteins (G$_{olf}$ and G$_{i3}$) and two Bouvier specific isoforms (G$_{oA}$ and G$_{oB}$) of the Go protein that spring from the same gene (GNAO1). Inoue et al. G$_q$ chimera replacing the six C-terminal amino acids have identical sequences for G$_{i1-2}$, G$_{oA-B}$, and G$_q$ and G$_{11}$. All analyses herein used the 11 common G proteins, a G$_o$ average of the G$_{oA}$ and G$_{oB}$ isoforms and left out G$_{olf}$ and G$_{i3}$ (not present in Bouvier et al.) while identical chimeras from Inoue et al. were used to represent the both members of the pairs G$_{i1-2}$, G$_{oA-B}$, and G$_q$ and G$_{11}$, respectively. Abbreviations: wt: wildtype; RlucII: Renilla luciferase 2.

| Family | G protein (protein) | Gene name | UniProt name | UniProt identifier | Bouvier | Inoue |
|---|---|---|---|---|---|---|
| G$_s$ | G$_s$ | GNAS | GNAS2 | P63092 | wt +RlucII at pos 67 | G$_{q1-353}$-RQYELL |
| | G$_{olf}$ | GNAL | GNAL | P38405 | - | G$_{q1-353}$-KQYELL |
| G$_{i/o}$ | G$_{i1}$ | GNAI1 | GNAI1 | P63096-1 | wt | |
| | G$_{i2}$ | GNAI2 | GNAI2 | P04899-1 | wt | G$_{q1-353}$- KDCGLF |
| | G$_{i3}$ | GNAI3 | GNAI3 | P08754 | - | G$_{q1-353}$-KDCGLY |
| | G$_{oA}$ | GNAO1 | GNAO | P09471-1 | wt | |
| | G$_{oB}$ | GNAO1 | GNAO | P09471-2 | wt | G$_{q1-353}$-RGCGLY |
| | G$_z$ | GNAZ | GNAZ | P19086-1 | wt | G$_{q1-353}$-KYIGLC |
| | G$_{t1}$ (transducin) | GNAT1 | GNAT1 | P11488 | - | |
| | G$_{t2}$ (transducin) | GNAT2 | GNAT2 | P19087 | - | Identical to G$_{q1-353}$-G$_{i1-2}$ chimera |
| | G$_{gust}$ (gustducin) | GNAT3 | GNAT3 | A8MTJ3 | - | |
| G$_{q/11}$ | G$_q$ | GNAQ | GNAQ | P50148-1 | wt | wt |
| | G$_{11}$ | GNA11 | GNA11 | P29992-1 | wt | G$_{q1-353}$-KEYNLV |
| | G$_{14}$ | GNA14 | GNA14 | O95837-1 | wt | G$_{q1-353}$-REFNLV |
| | G$_{15}$ | GNA15 | GNA15 | P30679-1 | wt | G$_{q1-353}$-DEINLL |
| G$_{12/13}$ | G$_{12}$ | GNA12 | GNA12 | Q03113-1 | wt | G$_{q1-353}$-KDIMLQ |
| | G$_{13}$ | GNA13 | GNA13 | Q14344-1 | wt | G$_{q1-353}$- KQLMLQ |

*supplement 1*). These findings define a sizeable reference set of consensus G protein couplings and show that consistent large-scale profiling studies generate more comparable results than literature.

## Common coupling map unifies GPCR-G protein datasets and opens analysis

To enable quantitative correlation of the Bouvier, Inoue, and future couplings, we established a data processing protocol giving the highest similarity and correlation of coupling measurements across datasets (Materials and methods). This uses log transformed EC$_{50}$ values and minimum-maximum normalized E$_{max}$ values combined into a unified log(E$_{max}$/EC$_{50}$) value and an aggregation of G proteins onto families based on the maximum subtype values (*Figure 2A*). Based on this protocol, we created a common coupling map, integrating the Bouvier, Inoue, and GtP datasets (*Figure 2B*). This unified coupling map establishes that it is possible to obtain comparable quantitative values despite the differences between biosensors and enables quantitative cross-study comparisons herein and in future studies from the field.

To enable any researcher to use the coupling map, we have availed a 'G protein couplings' browser (https://gproteindb.org/signprot/couplings) in GproteinDb (*Pándy-Szekeres et al., 2022*), which is a separate database but also accessible via the GPCRdb web site (*Kooistra et al., 2021*). By default, the coupling map (*Figure 2B*) and browser only show 'supported' couplings with evidence from two datasets, but there is an option (first blue button) to change the level of support to only one (for most complete coverage of GPCRs) or to three (for the highest confidence) sources. As an exception, GtP couplings do not require additional support, as they are in most cases supported by multiple independent publications. We propose a standardized terminology to describe couplings based on their level

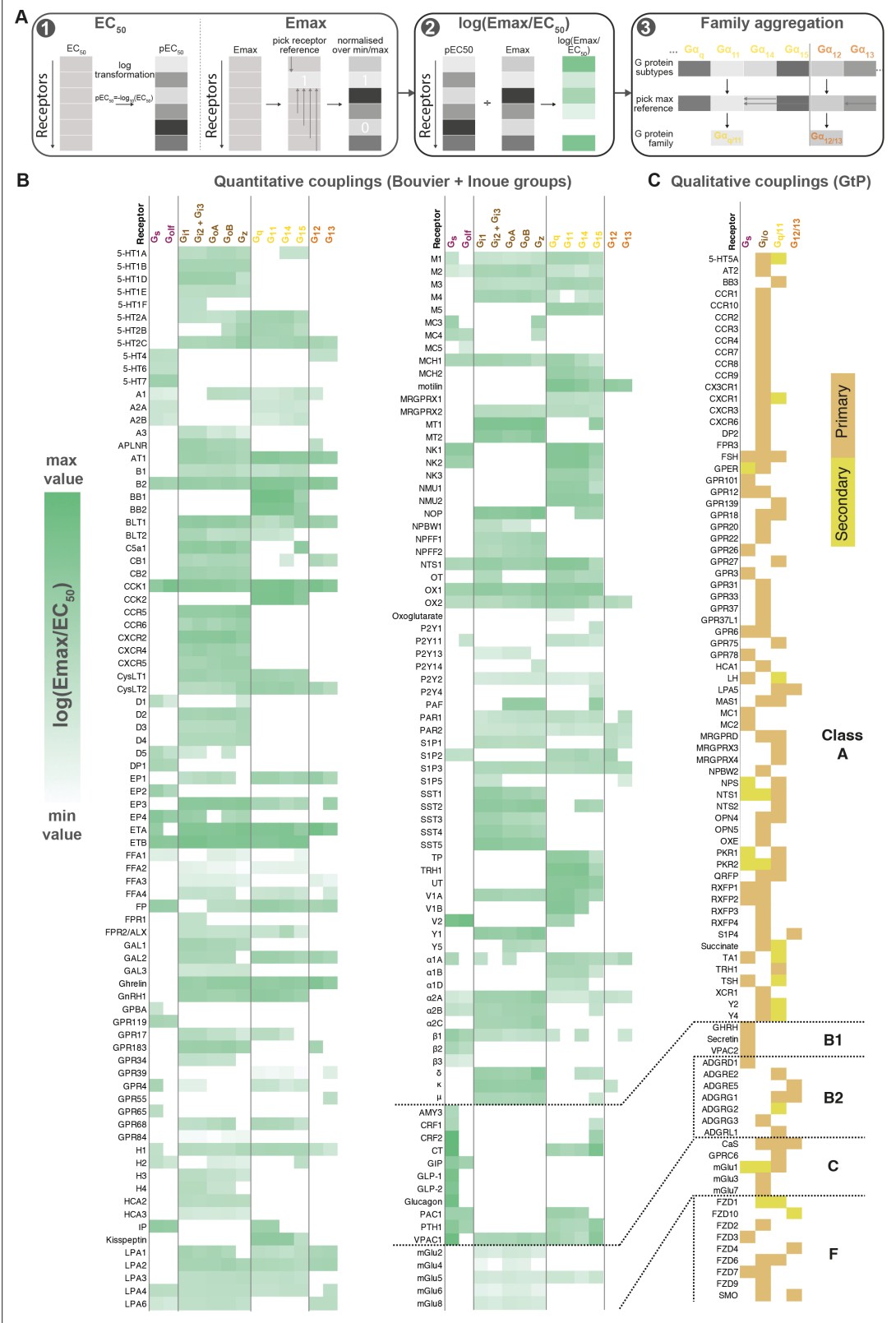

**Figure 2.** Map of G protein-coupled receptor (GPCR-G) protein couplings supported by at least two studies. (**A**) Normalization approach. $E_{max}$ minimum-maximum normalization, $EC_{50}$ log transformation and use of $\log(E_{max}/EC_{50})$ as a combined measure with member-to-family aggregation by maximum G protein value. (**B**) Heatmap representation of $\log(E_{max}/EC_{50})$ values for 166 GPCRs tested by the Bouvier and/or Inoue labs (for couplings with dual data sources a mean is used). For Guide to Pharmacology (GtP), a G protein subtype is considered supported if the family has a known

coupling. $G_{i2}$ and $G_{i3}$ are represented by the same/identical chimeric G protein in Inoue's dataset (**Table 1**). (**C**) Heatmap representation of primary and secondary transducers for 90 GPCRs which couplings are only covered by the GtP database. (**B–C**) Empty cells (white) indicate no coupling. All source values are available in tab 'Fig_4' in **Source data 5**. Note: Researchers wishing to use this coupling map, optionally after applying own reliability criteria or cut-offs, can do so for any set of couplings in GproteinDb (**Pándy-Szekeres et al., 2022**).

**Table 2.** Terms and definitions used to classify G protein-coupled receptor (GPCR-G) protein couplings.

| Term | Definition |
| --- | --- |
| Supported | Coupling or noncoupling datapoints that are supported by at least one other dataset, that is, at least two in total. An exception is made for couplings reported only in the GtP database, and not yet tested in a quantitative dataset, as the couplings in GtP are in most cases supported by multiple independent publications. |
| Novel | Coupling that is supported by Bouvier and Inoue but not present in GtP. |
| Proposed | Coupling identified in one but not yet tested in a second quantitative dataset (here Bouvier or Inoue). This refers to mainly receptors, but also G protein subtypes, that have only been tested in one quantitative dataset. Couplings reported only in the GtP database are not considered 'proposed' but 'supported' because they in most cases are based on the annotation of multiple independent publications. |
| Unique | Coupling that is unique in one dataset (other datasets have no coupling). |
| Missing | Noncoupling in the given dataset but coupling in the other compared datasets. |

GtP: Guide to Pharmacology.

of experimental support from independent groups (**Table 2**). The criterion of supporting independent data, and the terms 'proposed' and 'supported', are already used by the Nomenclature Committee of IUPHAR for GPCR deorphanization. Furthermore, the online coupling browser allows any researcher to use only a subset of datasets, or to apply filters to the Log($E_{max}/EC_{50}$), $E_{max}$, and $EC_{50}$ values. Finally, users can filter datapoints based on a statistical reliability score in the form of the number of SDs from basal response.

## Research advances – insights on GPCR-G protein selectivity

### The Bouvier and Inoue datasets jointly support 101 novel G protein couplings

We next identified the 'novel' G protein couplings for which a family annotation is missing in GtP but have high confidence from dual support by the Bouvier and Inoue groups (**Figure 1—figure supplement 1**). This revealed 38 receptors with novel couplings to 101 G proteins distributed across all families: $G_s$: 4, $G_{i/o}$: 15, $G_{q/11}$: 10, and $G_{12/13}$: 21 (**Figure 3**). The largest expansions – an increase by three of G protein families – were obtained for the histamine $H_1$ and endothelin $ET_A$ receptors which were found to couple to all G protein families but only have $G_{q/11}$-coupling in GtP. Whereas it could be expected that GtP would miss couplings, we also analyzed if its expert curation excluded couplings that may be false positives as they are contradicted by both quantitative studies. This uncovered such $G_s$-coupling to the $\alpha_{2C}$-adrenoceptor and cannabinoid $CB_{1-2}$ receptors, $G_{12/13}$ coupling to the purinergic $P2Y_2$ receptor and $G_{i/o}$-coupling to the $\beta_2$-adrenoceptor, which however had weak $G_z$ and $G_{oB}$ coupling in the Bouvier study but did not cross the signal threshold (**Avet et al., 2020**). Notably, this is only 2% (5/254) of all GtP's GPCR-G protein family pairs. Taken together, these findings serve to quantify the expansion of the GPCR-G protein 'couplome' while also confirming the outstanding accuracy of the expert annotation in the GtP database. Therefore, the large number of new couplings is mainly due to missing investigations in literature rather than incorrect curation.

### Half of GPCRs are selective for a single whereas 5% promiscuously activate all G protein families

To gain insight into their levels of coupling selectivity, we intersected the G protein profiles of all receptors and counted the number of coupling partners for GPCRs and G proteins (**Figure 4**). On

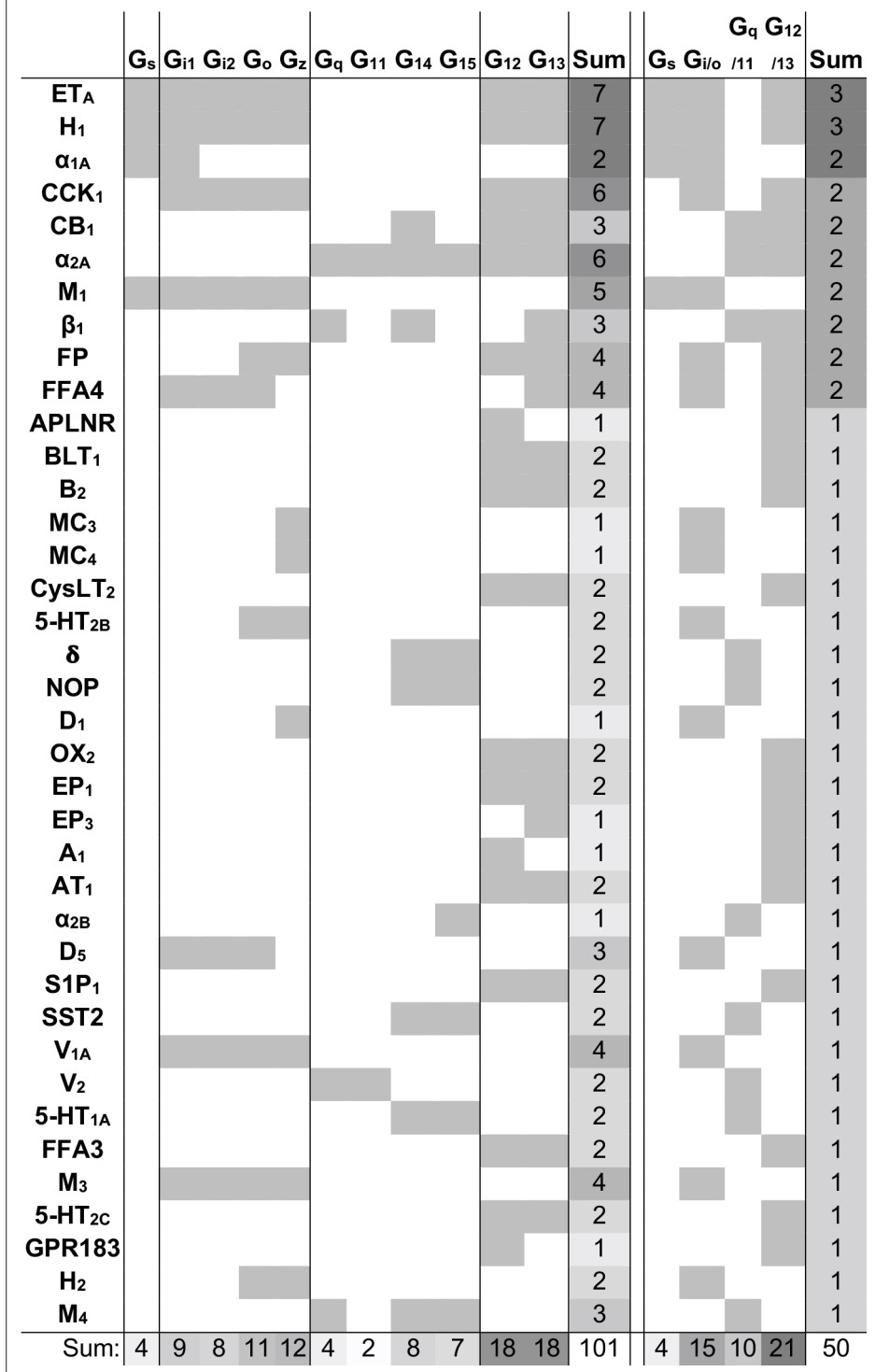

**Figure 3.** Novel G protein couplings identified by both the Bouvier and Inoue groups. Novel couplings identified by the Bouvier and Inoue groups but not in Guide to Pharmacology. G protein families are here considered shared if at least one specific subtype is found to couple in both dataset, which is not the case for the bottommost four receptors.

the G protein family level (topmost in *Figure 4*), our analysis spans 90 receptors with data only in GtP and 166 GPCRs with data from the Bouvier and/or Inoue groups – totaling 256 receptors. We require couplings from the Bouvier and Inoue groups to be supported by a second dataset (GtP couplings are already typically supported by several publications). We find that these 256 GPCRs couple to on average 1.7 G protein families distributed as 126 single-, 83 double-, 34 triple- and 13 all-family activating receptors. The share of fully selective (single-family activating) receptors differs largely across G protein families spanning from 6% for $G_{12/13}$ to 22% for $G_{q/11}$, 26% for $G_s$ and up to 40% for $G_{i/o}$ (tab 'Fig_4' in *Source data 5*).

Interestingly, all fully promiscuous receptors are class A GPCRs: adenosine $A_1$, adrenergic $\alpha_{1A,2A}$ and $\beta_1$, bradykinin $B_2$, cannabinoid $CB_1$, cholecystokinin $CCK_1$, endothelin $ET_A$, prostanoid FP, GPR4, histamine $H_1$, lysophospholipid $LPA_4$, and orexin $OX_2$ receptors (216 class A, 11 class B1, and 5 class C GPCRs have been profiled, so far). Conversely, a G protein family has supported couplings to on average 112 GPCRs (28% of all receptors) distributed as $G_s$: 87, $G_{i/o}$: 176, $G_{q/11}$: 134, and $G_{12/13}$: 49 receptors (34%, 69%, 52%, and 19%, respectively of all receptors, *Figure 4C*). Given that 101 of the GPCR-G protein family pairs tested by the Bouvier and Inoue groups represent novel couplings (above), more couplings are expected to be identified as expanding and confirmatory studies emerge. Hence, whereas the results described here represent the currently known supported couplings, the total 'couplome' will undoubtedly comprise additional yet undetected and unconfirmed couplings, especially among receptors never profiled with a pan-G protein platform.

## Three-quarters of GPCR activate all G proteins belonging to the same family

Within each G protein family (rows 2–5 in *Figure 4*), on average 73% of GPCRs promiscuously activate all its members ($G_{q/11}$: 73%, $G_{12/13}$: 66%, $G_{i/o}$: 75%, and $G_s$: 80%). In contrast, activation of only one subtype of a G protein family is only observed for 11 $G_s$, 4 $G_z$, 1 $G_{14}$, 10 $G_{15}$, 5 $G_{12}$, and 10 $G_{13}$-coupled receptors (*Figure 2* or *Source data 5*). Most other receptors activate a subset of G proteins in each family. Strikingly, $P2Y_{1,4}$ ($G_{15}$), $P2Y_{14}$ ($G_z$), and GPR55 ($G_{13}$) are fully selective also when considering G proteins from all families, that is, they only couple to a single of the 16 human G proteins (1.4 × SD cut-off applied, 6 $G_s$-coupling receptors are left out, as $G_{olf}$ has so far only been tested by the Inoue group). However, the three purinergic receptors have additional couplings although not supported by a second dataset ($P2Y_4$ and $P2Y_{14}$) or above the 1.4 × SD cut-off ($P2Y_1$ was below), and it is possible that as the characterizations expand even fewer, or no receptors are found to engage only a single G protein.

The abundant coupling of $G_{i/o}$ and $G_{q/11}$ to many receptors with dual (or more) pathways suggests that these G proteins often have more versatile functions. In all, our meta-analysis of GPCR-G protein selectivity points to intriguing differences where some receptors selectively signal via a single effector whereas other GPCRs promiscuously activate all four G protein pathways. Such selective or combined profiles, in interplay with differential spatiotemporal expression (*Avet et al., 2020*), can be critical to achieve a specific physiological effect.

## G protein co-coupling reflects phylogeny, but all G protein families have an odd member

To assess whether the evolutionary classification dividing G proteins into four families is also representative of their pharmacology, we investigated the correlated coupling of G proteins to receptors. The overall correlation was assessed with Pearson standard correlation coefficients (*Figure 5A–B*) and broken down into shared coupling/noncoupling (from Jaccard indices, *Figure 5C*) and activation levels mean log($E_{max}/EC_{50}$, *Figure 5D*). We find that all three comparisons show the strongest correlations for G proteins belonging to the same G protein family (boxed in *Figure 5A and C–D*). This demonstrates that the pharmacological relationships, that is, the coupling selectivity and activation level, do indeed reflect the phylogenetic relationships. This is important, as this is how the human G proteins into four families have been classified traditionally, but it had not been tested if this classification applies to pharmacological profiles of the scales analyzed herein and likely increasingly more common in future studies.

Several G protein pairs stand out with exceptionally high overall correlated coupling (Pearson standard correlation coefficient): $G_{i1}$-$G_{i2}$ (0.99), $G_{oA}$-$G_{oB}$ (0.96), and $G_q$-$G_{11}$ (0.99). Interestingly, the correlated coupling is lower for the only pairs within the $G_s$ and $G_{12/13}$ families, $G_s$-$G_{olf}$ (0.77) and $G_{12}$-$G_{13}$

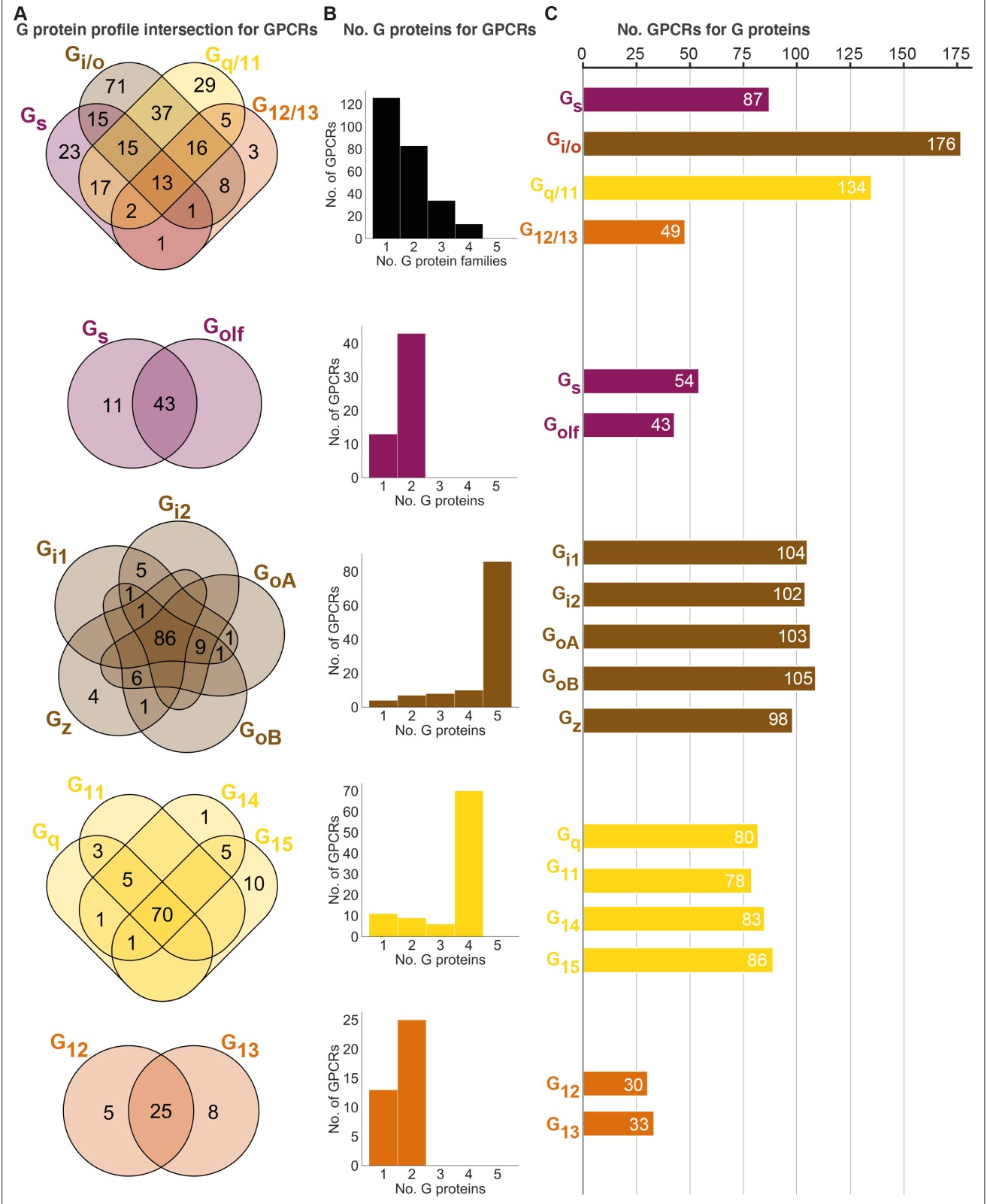

**Figure 4.** Unifying the three datasets reveals a large diversity in G protein-coupled receptor (GPCR) coupling selectivity. (**A**) GPCR-G protein selectivity on the G protein family (top) and subtype levels: Venn diagrams showing the numbers of shared and unique receptors. The 0 values (no receptors) are omitted for clarity. (**B**) Receptor coupling promiscuity: number of receptors that couple to 1–4 G protein families (top) and 1–5 subtypes. (**C**) G protein coupling promiscuity: number of receptors that couple to each G protein family (top) or subtype. (**A–C**) Panels **B–C** are based on the couplings from the

*Figure 4 continued on next page*

*Figure 4 continued*

Bouvier and Inoue groups that are also supported by a second dataset and panel **A** additionally includes GtP couplings. This analysis of the $G_s$ family leaves out 11 receptors tested for coupling to $G_s$ but not to $G_{olf}$, and $G_{olf}$ couplings are only counted if there is a supported $G_s$ coupling. All source data are available in tab 'Fig_4' in *Source data 5*.

(0.79), respectively. Furthermore, both $G_{i/o}$ and $G_{q/11}$ have an 'odd' member, $G_z$ and $G_{15}$, respectively (mean of 0.78 and 0.76 to other family members, respectively). The fact that all G protein families have an odd member shows that all signaling pathways have a transducer toolbox allowing them to differentiate signaling.

## $G_s$ or $G_{i/o}$ coupling is selective while $G_{12/13}$ is promiscuously activated with $G_{i/o}$ and $G_{q/11}$

The $G_s$ and $G_{i/o}$ families have an inverse correlation in all of the overall, coupling/noncoupling, and activation level comparisons (darkest red in *Figure 5A and C–D*). This means that $G_s$ and $G_{i/o}$ rarely co-couple to GPCRs and, when they do, they do so with a large difference in their strength of activation. This inverse correlation where only one or the other G protein is activated is in agreement with their function, as $G_s$ stimulates, and $G_{i/o}$ inhibits production of the same cellular second messenger, cAMP. This allows opposite physiological responses to be mediated with temporal selectivity in the same cell by activating different receptors at different times. Differential engagement of the $G_{i/o}$ family could also have functional consequences through other pathways, as $G_i$ was recently shown to be required for scaffolding β-arrestin binding and signaling for some receptors (*Smith et al., 2021*).

For $G_{12/13}$, we instead find a positive correlation with the $G_{i/o}$ and $G_{q/11}$ families. This is in agreement with *Figure 4* (first Venn) which shows that only 3 GPCRs couple to only $G_{12/13}$ and 3 couple to the rare $G_{12/13}$-$G_s$ family pair, whereas the vast majority of GPCRs, 38 couple to $G_{12/13}$ also couple to the $G_{i/o}$ and/or $G_{q/11}$ family. Furthermore, we find an intriguing unique selectivity mechanism for the $G_{i/o}$ and $G_{q/11}$ families. These families have the most frequent co-coupling (light blue in *Figure 5C*) but have a high difference in average activation levels (red in *Figure 5D*). Together, this shows that selective binding of only $G_{i/o}$ or $G_{q/11}$ is uncommon, but selectivity can be achieved by differential levels of activation. This possibility to achieve selectivity via differential activation is likely important, as many GPCRs, 81 couple to both the $G_{i/o}$ and $G_{q/11}$ families. Their high difference in activation levels allows for a selective activation, which could shift upon the presentation of alternative endogenous or surrogate ligands with a signaling bias. Furthermore, it should be pointed out that many reported dual couplings to $G_i$ and $G_q$ may be because phospholipase c (PLC), a direct effector of $G_q$, can also be activated by $G_{\beta\gamma}$ from $G_i$ (i.e. sensitive to the $G_i$ inhibitor pertussis toxin). Thus, the concept of potential scaffolding of PLC by $G_q$ may be required for $G_i$-derived $G_{\beta\gamma}$ activation (*Pfeil et al., 2020*).

## Differential tissue expression gives G proteins in the same family large spatial selectivity

To study how tissue expression may influence G protein selectivity, we analyzed consensus transcript expression levels for 50 tissues and 16 organs (here aggregated into 8) from the Human Protein Atlas (HPA) (*Uhlén et al., 2015*), which also includes data from the Genotype-Tissue Expression (GTeX) project (*Lonsdale et al., 2013*; *Figure 6*). We find that the most ubiquitously expressed G proteins are $G_s$ and $G_{i2}$ which are expressed in all 50 tissues and all organ categories at a level that is within the first quartile of all normalized transcripts per million (nTPM) values. In contrast, $G_{i3}$, $G_{t1}$, $G_{t2}$, $G_{t3}$, and $G_{14}$ have none or only one tissue with a first quartile expression. To facilitate comparison across G proteins, we employed HPA's nTPM values after z-score transformation of each G protein across tissues. A z-transformation of the same G proteins visualizing the relative tissue levels of expression, shows a very focused expression of the transducins, $G_{t1}$ and $G_{t2}$, in the retina and gustducin $G_{t3}$ in the gastrointestinal tract reflecting their role in vision and taste, respectively (*Figure 6—figure supplement 1A*). Notably, each G protein family has at least one subtype that is preferentially expressed in the brain: $G_s$: $G_{olf}$, $G_{i/o}$: $G_o$ and $G_z$, $G_{q/11}$: $G_q$, and $G_{12/13}$: $G_{12}$ (*Figure 6* and *Figure 6—figure supplement 1A*).

Given that the tissue expression varies largely for G proteins, we next sought to determine to what extent their tissue expression correlations differ from the traditional four G protein families

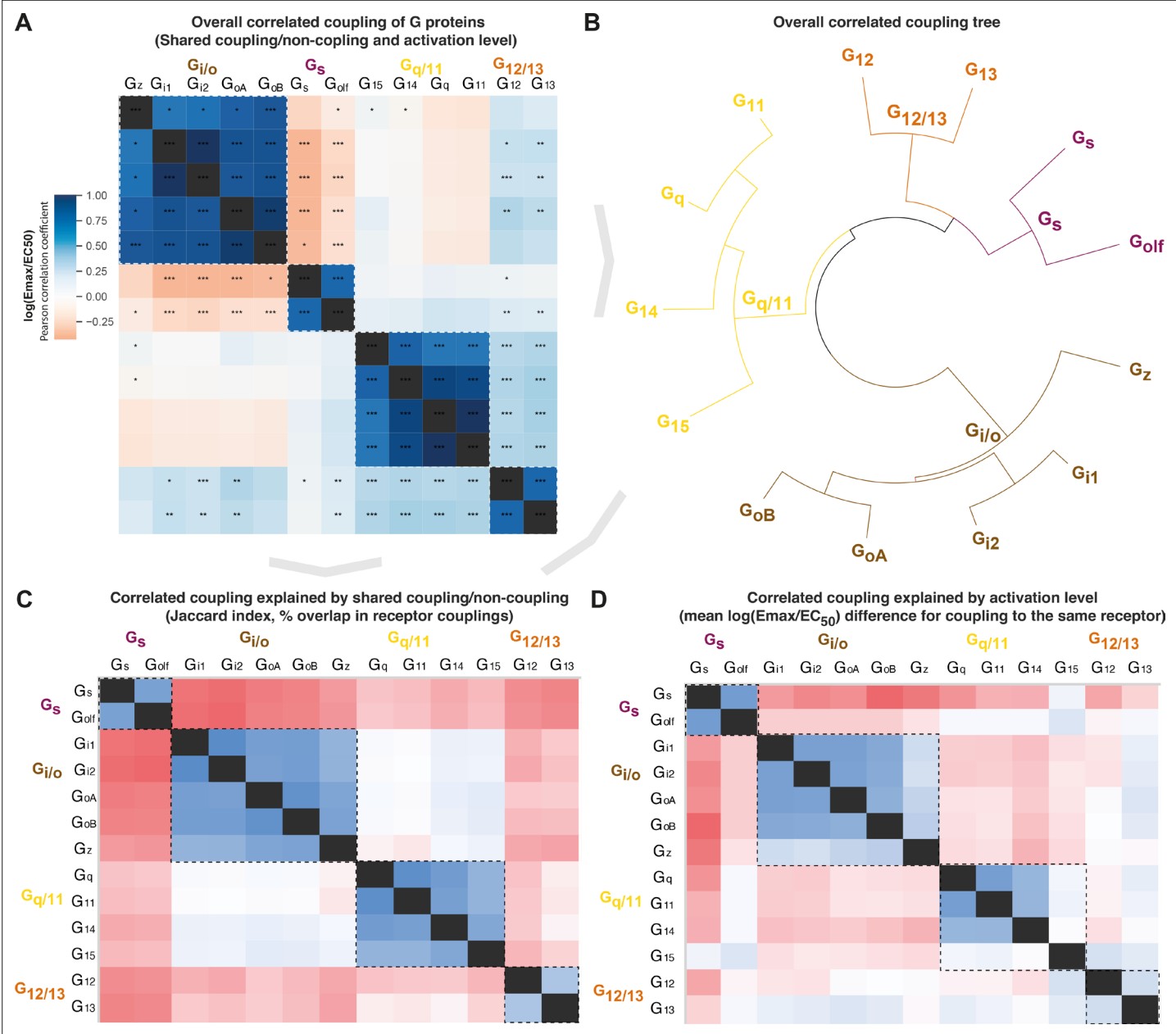

**Figure 5.** Correlated coupling of G proteins based on their receptor profiles. (**A**) Overall correlated coupling of G proteins quantified by the Pearson standard correlation coefficient, which gives a measure of the strength of the linear relationship between two G proteins the $\log(E_{max}/EC_{50})$ value of noncoupling datapoints was set to 0. Statistically significant pairwise correlations are indicated in cells by *p≤0.05; **p≤0.005, and ***p≤0.0005. (**B**) Overall coupling correlation of G proteins shown as a tree. (**C**) Correlated coupling explained by shared coupling/noncoupling quantified as Jaccard indices (% of couplings to the same G protein-coupled receptor [GPCRs]). (**D**) Correlated coupling explained by activation level quantified as the differences in average $\log(E_{max}/EC_{50})$ when a G protein pair couples to the same receptor. The values are averages of the $\log(E_{max}/EC_{50})$ averages of Bouvier and Inoue values, except where data is only available in one dataset, that is, Bouvier only: $G_{i1}$-$G_{l2}$, $G_{oA}$-$G_{oB}$, $G_q$-$G_{11}$, and Inoue only: $G_{olf}$~all G proteins. (**A, C–D**) All G protein couplings are for class A GPCRs and supported by two datasets (Bouvier or Inoue group or Guide to Pharmacology), and their source values are available in tab 'Fig_5' in *Source data 5*. (**A–D**) For G proteins (all but $G_{olf}$) and receptors tested by both the Bouvier and Inoue groups, an average of averages from the two groups is used. The $G_{oA}$ and $G_{oB}$ couplings from Bouvier were compared to the $G_o$ values from Inoue which does not distinguish isoforms.

that represent phylogeny and GPCR coupling (*Figure 5*) relationships. To this end, we calculated the co-expression for each G protein pair using Pearson standard correlation coefficients (*Figure 6—figure supplement 1B*). Notably, we find that all four G protein families have members that group apart in the dendrogram (the two members of the $G_s$ family, $G_s$ and $G_{olf}$, do not cluster adjacently but

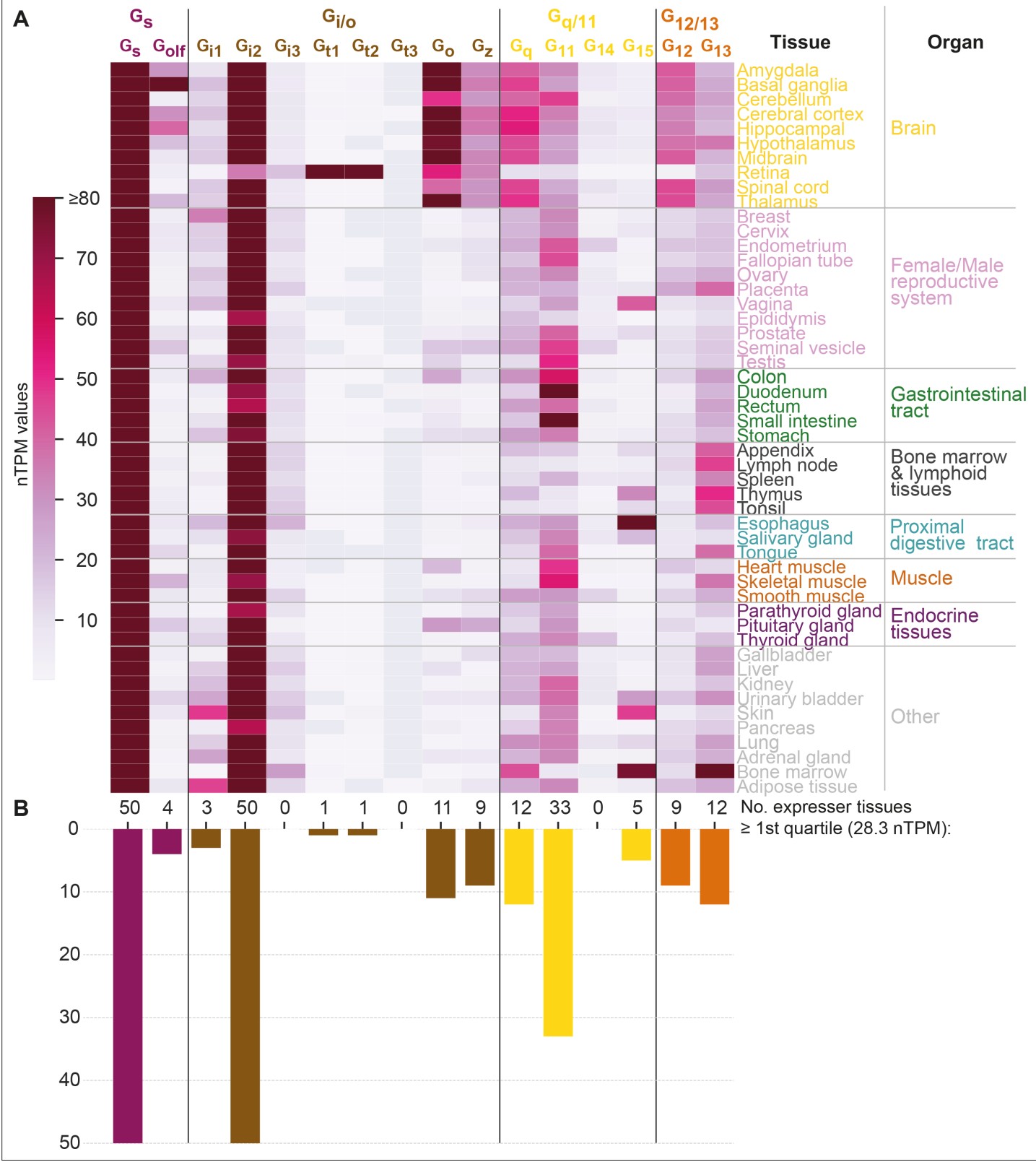

**Figure 6.** G protein tissue and organ expression profiles. (**A**) Expression heatmap of all 16 human G proteins across 50 tissues and 16 organs (here grouped in 8 categories) extracted from the Human Protein Atlas (***Uhlén et al., 2015***), which also includes data from the Genotype-Tissue Expression project (***Lonsdale et al., 2013***). The coloring denotes the normalized transcripts per million (nTPM) for each G protein and tissue capped at the median

*Figure 6 continued on next page*

*Figure 6 continued*

of all maximum G protein expression values ($G_{t3}$ at 80 nTPM in retina). (**B**) Number of tissues for which the given G protein has an expression ≥ the first quartile expression threshold (28.4 nTPM) across all 16 G proteins and 50 tissues.

The online version of this article includes the following figure supplement(s) for figure 6:

**Figure supplement 1.** Correlated expression of G proteins.

---

not as a pair). Instead, we find a coherent cluster for the G proteins that are predominantly expressed in the brain, $G_{olf}$, $G_o$, $G_z$, $G_q$, and $G_{12}$ and cover all four families. Furthermore, among the remaining co-expressing G proteins, three out of the four closest pairs also span families ($G_{i3}$-$G_{15}$, $G_{i2}$-$G_{13}$, and $G_{t3}$-$G_{11}$ but not $G_{t1}$-$G_{t2}$). Taken together, this shows that there is no overall clustering of subtypes within the G protein families, which instead exploit differential expression to gain spatial selectivity. For example, in the $G_{i/o}$ family $G_o$ and $G_z$ are restricted to brain regions, while the other subtypes have four different peripheral profiles.

## Technological considerations – biosensor sensitivity

### Bouvier and Inoue biosensors appear more sensitive for $G_{15}$ and, $G_s$ and $G_{12}$, respectively

To evaluate the sensitivity of the Bouvier and Inoue biosensors for different G proteins, we compared their observed couplings for the 70 common receptors to identify 'unique' and 'missing' couplings (defined in *Table 2*). For $G_{15}$, Bouvier reported 30% unique couplings, several of which were validated in $Ca^{2+}$ assays (*Avet et al., 2020*), whereas Inoue instead misses 13% of couplings present in GtP (*Figure 7*). This indicates that $G_{15}$ couplings are underrepresented in Inoue's data and in literature (annotated in GtP). The few published couplings for $G_{15}$ are likely explained by its lack of expression in the cells most used for in vitro experiments, such as HEK293 cells (*Avet et al., 2020*), the lack until recently of enough sensitive sensors to directly measuring $G_{15}$ activity, and by the lack or weak (at very high concentration) effect on this subtype by $G_{q/11}$ inhibitor tools YM-254890 and FR900359, respectively (*Malfacini et al., 2019*).

For $G_s$ and $G_{12}$, 27% and 24%, respectively of Inoue's couplings are unique, whereas Bouvier instead misses 7% of $G_s$ and 6% of $G_{12}$ couplings in GtP. These two G proteins have smaller assay windows

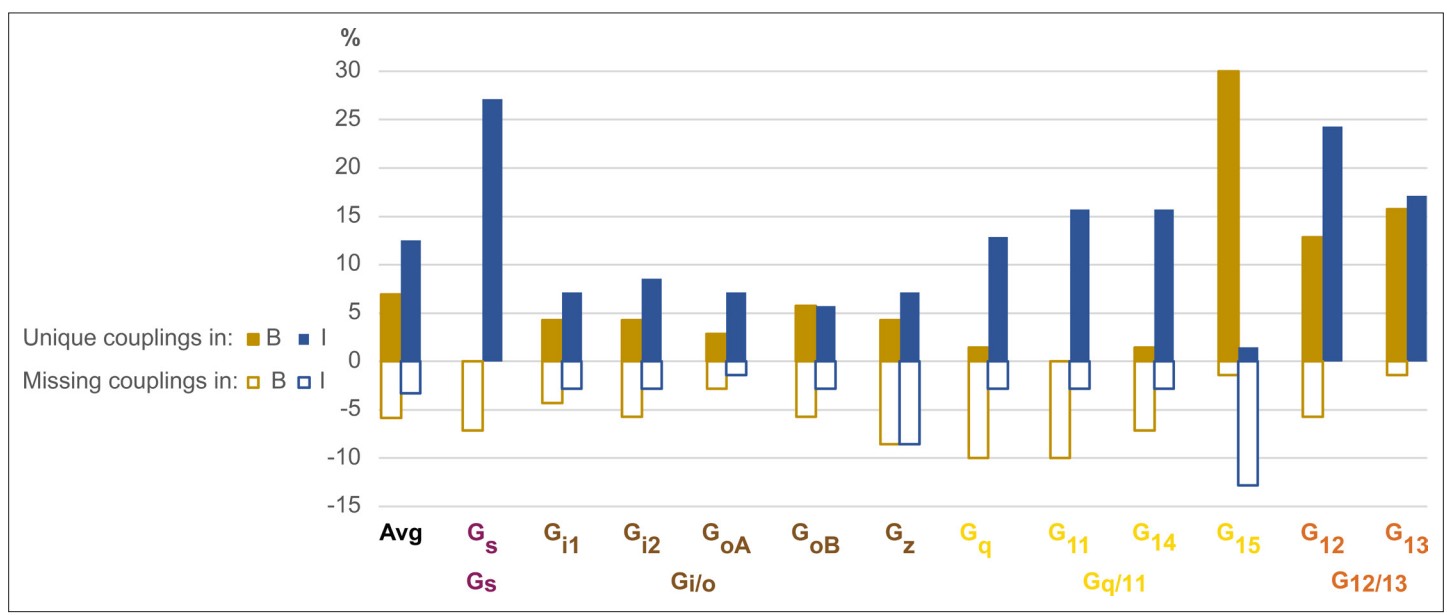

**Figure 7.** Unique and missing couplings in the Bouvier and Inoue datasets. Unique and missing couplings (defined in *Table 2*) among all 70 receptors common to Bouvier, Inoue, and Guide to Pharmacology (GtP) (the 100% includes noncoupling receptors). Missing couplings are shown as negative values. For GtP, we consider coupling to a G protein subtype possible if a coupling has been observed for the respective family. Unique couplings are hidden by default in the online G protein couplings browser in GproteinDb, as they await the independent support by a second group (*Pándy-Szekeres et al., 2022*).

which may explain their lower representation in the Bouvier dataset (tab 'DataStats' in *Source data 1*). Furthermore, for $G_q$, $G_{11}$, and $G_{14}$, 13%, 16%, and 16%, respectively of Inoue's couplings unique. In contrast, Bouvier is missing 10%, 10%, and 7%, respectively of the known couplings in GtP which indicates an underrepresentation or alternatively that literature has mistaken $G_i$-derived βγ activation of the PLC-IP3-calcium signaling pathway for $G_q$-mediated signaling (*Blank et al., 1991*; *Tomura et al., 1997*; *Jiang et al., 1996*; *Camps et al., 1992*). Notably, although Inoue used the wildtype protein for $G_q$, it has a similar or higher frequency of unique couplings compared to all other measured G proteins ($G_q$: 13% vs. $G_{i1}$: 7%, $G_{i2}$: 9%, $G_{oA}$: 7%, $G_{oB}$: 6%, $G_z$: 13%, $G_{11}$: 16%, $G_{14}$: 16%, $G_{15}$: 1% and $G_{13}$: 17%), except $G_s$ and $G_{12}$. This suggests that the many unique couplings in Inoue's dataset may be due to high sensitivity rather than the use of chimeric G proteins carrying over of $G_q$ coupling to the other G proteins.

To investigate how unique couplings are affected by weak couplings and biosensor sensitivity, we compared their efficacies and potencies (rows 106–110 in tab 'BIG-QualComp' in *Source data 3*). We find that the average $E_{max}$ values are 18% and 15% lower for unique than for supported couplings in the Bouvier and Inoue datasets, respectively. Furthermore, their average $pEC_{50}$ values are 0.4 and 1.5 log units – 2.5-fold and 32-fold, respectively – lower than the supported couplings. This indicates that a part of the differences across datasets is due to weak couplings that can only be detected in the most sensitive assays for the given G proteins and is difficult to distinguish from basal levels (this study used a cut-off of $E_{max}$ >1.4 SDs over basal response). However, couplings may also be overrepresented and determining couplings definitely would require further studies employing independent ('orthogonal') high-sensitivity biosensors. Until then, we recommend the requirement of two independent studies to support of a GPCR-G protein coupling, as is the criterion used herein for the novel couplings and the default in the online G protein couplings resource (*Armstrong et al., 2020*).

## Discussion

Given that researchers are now faced with three large G protein coupling datasets varying in coverage and couplings, we established a common normalization protocol making quantified values comparable across data sources and a common coupling map. Previous analyses of GPCR-G protein selectivity have all been based on a single dataset from the GtP (*Flock et al., 2017*) database or the Inoue (*Inoue et al., 2019*) or Bouvier (*Avet et al., 2020*) laboratories, whereas the analysis herein combined these datasets to establish couplings supported by two studies (including GtP annotations of multiple literature reports) for a total of 256 GPCRs. This provides a sizeable and reliable reference dataset of supported quantitative couplings suitable for any study.

By analyzing the common coupling map, we find that 101 novel couplings are supported by both the Bouvier and Inoue datasets but were not known in GtP which annotates coupling from the literature. We also find that very few receptors, 13 (5%) promiscuously couple to all four G protein families whereas the receptor numbers instead increase several-fold along with selectivity: 34 triple-, 83 double-, and 126 single-family coupling receptors (*Figure 4B*). As more large coupling datasets become available, this is likely to shift somewhat from less selectivity toward higher promiscuity. Furthermore, the finding that on average only 27% of GPCRs activate only a subset of the G proteins in each family (*Figure 4*) opens the possibility that signaling bias on the level of specific subtypes may be much more frequent than currently appreciated. G proteins that belong to the same family can have diverse functional outcomes pertaining to effector engagement selectivity and kinetic profiles (*Jiang and Bajpayee, 2009*; *Ho and Wong, 2001*), suggesting that bias within a G protein family may have physiological and/or therapeutic implications (*Anderson et al., 2020*). The observation that many GPCRs can couple to more than one family of G proteins but still can show selectivity between members of a same family, opens important questions concerning the structural determinants of such multifaceted selectivity profiles.

The GPCR-G protein couplings analyzed herein represent the overall couplings that are possible, but may differ largely physiologically across specific tissues. Our analysis showed a tremendous span in how many tissues that express a given G protein. Some G proteins, including the transducins ($G_{t1}$ and $G_{t2}$) and gustducin ($G_{t3}$) have a restricted expression in mainly one tissue (retina and gastrointestinal tract, respectively), whereas the most ubiquitously expressed G proteins, $G_s$ and $G_{i2}$ are highly expressed in all 50 analyzed tissues. Notably, G proteins belonging to the same family differ largely in their tissue profiles, for example, each G protein family has a single or two subtypes that is/

are preferentially expressed in the brain. Given that the activity (here $\log(E_{max}/EC_{50})$ values) can vary substantially across G protein family subtypes (*Figure 2*), this provides cells with a mechanism to shift the system bias toward different intracellular signaling pathways. Such system bias can be modulated further through differential expression of also the receptors and multiple downstream intracellular effectors (*Kolb et al., 2022*). A recent study found a strong spatial correlation of summed mRNA expression levels for $G_s$-, $G_{i/o}$-, and $G_{q/11}$-linked receptors in humans, macaques, and mice (*Treviño and Manjarrez, 2022*). This suggests that the expression patterns of receptors linked to these major G protein families are strongly interdependent and could therefore act together to balance each other in specific tissues. Future studies would be needed to profile the couplings of natively expressed GPCRs and G proteins across different cell types. Meanwhile, a strategy could be to analyze GPCR-G protein selectivity for a subset of G proteins that are selected based on the highest expression in the tissue of interest. For example, analysis of $G_{i/o}$-mediated signaling in the brain could focus on $G_o$ and $G_z$ based on their high expression, and the fact that $G_o$ and $G_z$ have quicker and slower nucleotide exchange, respectively, than other members of this family. Such separate meta-analysis of G proteins may be facilitated by the filtering options in the coupling maps in *Source data 5* and online in the G protein couplings browser in GproteinDb (*Pándy-Szekeres et al., 2022*).

The Inoue and Bouvier datasets represent the first steps to systematically characterize large G protein coupling profiles. Other laboratories may use other biosensors to expand the coverage of the coupling map, and our study facilitates this by indicating which of the Bouvier and Inoue biosensors that have the highest sensitivity for each G protein (*Figure 7*). However, future studies should also employ different biosensors and systems, as independent support is critical for validation and for distinguishing novel couplings from false positives among the many unique couplings (*Figure 7* and *Source data 3*). Of note, several biosensors have recently been published (*Olsen et al., 2020*; *Maziarz et al., 2020*; *Masuho et al., 2015*) (Gαβγ sensors first described in *Galés et al., 2005*; *Galés et al., 2006*; *Schrage et al., 2015*; *Breton et al., 2010*; *Bünemann et al., 2003*; *Janetopoulos et al., 2001*). Whereas a detailed comparison is out of the scope of this study, their pros and cons have recently been reviewed (*Wright and Bouvier, 2021*) and researchers should strive to use the most native proteins and expression, physiologically relevant tissues/cells (*Apostolakou et al., 2020*) and assays with sufficient sensitivity and window. Of note, most studies annotated in GtP have used downstream measurements, which are amplified, and may therefore differ from those obtained by biosensors that measure G protein activation (further discussed in *Avet et al., 2020*).

In all, our cross-dataset analysis has established a protocol and reference set aiding GPCR-G protein coupling studies. The selectivity profiles are the most comprehensive to date spanning 256 receptors with a very diverse activation of a single to all G protein pathways, and presented in a dedicated online G protein couplings browser in GproteinDb (*Flock et al., 2017*). The analyses and data presented herein will be very valuable to illuminate undercharacterized pharmacological phenomena such as constitutive activity (*Berg and Clarke, 2018*), precoupling of G proteins (*Galés et al., 2006*; *Civciristov et al., 2018*), and ligand-dependent biased G protein signaling (*Kenakin, 2019*), and to uncover their underlying determinants. They also present the foundation to integrate more coupling data as future studies expand the characterization of the 'couplome'.

# Materials and methods
## Study design

The primary objective of this study was to generate a unified map of G protein couplings (*Figure 2*) across the three available large datasets from the Bouvier (*Avet et al., 2020*) and Inoue (*Inoue et al., 2019*) groups, and the GtP database (*Armstrong et al., 2020*), respectively. To do this, we identified the $E_{max}$ SD cut-off, quantitative normalization protocol, and aggregation of G proteins into families giving the best possible agreement between the GPCR-G protein couplings from the Bouvier and Inoue groups. This analysis also involved assessing the agreement of the Bouvier and Inoue group datasets and determining the number of high-confidence novel couplings supported by these two datasets but not reported in the GtP database. Furthermore, it included a benchmarking of techniques to determine which biosensor produces the most reproducible qualitative coupling determination (coupling vs. noncoupling) for each G protein family. To this end, we compared the three datasets to pinpoint, for each dataset and G protein family, the fraction of couplings supported by another

dataset. The scientific analyses based on the map feature the most comprehensive analysis to date of GPCR-G protein selectivity. This was done by intersecting the 11 G proteins tested by both Bouvier and Inoue, within and across their families, with respect to the common and unique receptors that they couple to. Finally, the receptor profiles were also used to classify G proteins and to determine any co-correlation between different G protein subtypes and families to GPCRs.

## Coupling datasets

Updated spreadsheets containing the $pEC_{50}$, $E_{max}$, basal signals, and SD values were supplied by Asuka Inoue. Basal signals, spontaneous AP-TGF-α release (in % of total AP- TGF-α surface expression) were recalculated from raw data of the previous coupling-profiling campaign (*Inoue et al., 2019*) and their SD values were computed from independent experiments (n ≥3). This file contains the $NTS_1$ and $TRH_1$ receptors not included in the previous publication (*Inoue et al., 2019*). In the Inoue dataset, the protease-activated receptors PAR3-4 had negative $pEC_{50}$ values (concentration of mU/ml because the ligand, thrombin, was supplied with its enzymatic activity). For the easiness of integration into the coupling map, we added a value of 10 to their $pEC_{50}$ values. Data qualities (sigmoidal curves) for the individual GPCR-G protein pairs were manually inspected and concentration-response curves that did not converge nor exceed a threshold (typically, 3% AP-TGF-α release) were regarded as no activity.

The Bouvier dataset (n ≥3) contained some datapoints that were included or excluded based on dedicated analyses. First, we excluded ligand-promoted responses of overexpressed receptors that were equivalent to those of endogenously expressed receptors (yellow fill in tab 'B-EmEC' in *Source data 1*). Second, couplings with only approximate $E_{max}$ and $pEC_{50}$ values because the dose-response curve did not converge were only included if supported, and not contradicted, by the Inoue and/or GtP datasets (orange fill in tab 'B-EmEC' and analyzed separately in tab 'UnconvergedDRV' in *Source data 1*). This is because a coupling with an unprecise quantitative value is better than no coupling, especially when making qualitative comparisons (coupling vs. non-coupling). Based on these criteria seven couplings were included: $5-HT_{1D}-G_z$, $BLT_1-G_{11}$, $FFA3-G_{13}$, $GPR4-Gi_{1-2}$, $GPR84-G_{oA}$, and $κ-G_{12}$, and six couplings were excluded: $CCR5-G_{14}$, $CXCR5-G_q/G_{11}/_{G14}$, $GPR183-G_{13}$, and $κ-G_{13}$.

## Standard deviation cut-off

We made a special investigation of couplings that have a full dose-response curve but an $E_{max}$ less than two SDs from the basal signal (red fill in tab 'B-EmEC' and analyzed separately in tab tabs 'B- < 2SDs' and 'I- < 2SDs' in *Source data 1*). To achieve the best possible separation of putative false and real but weak couplings, we identified the threshold value (a number of SDs from basal signal) that gives the best agreement between the Bouvier and Inoue dataset among the common receptors and G proteins tested by both groups. The obtained cut-off, 1.4 SDs, was applied as a filter to exclude all Bouvier and Inoue $E_{max}$ and $pEC_{50}$ values for couplings below this cut-off (columns with ' >1.4 SDs' in the heading in tabs 'B-EmEC' and 'I-EmEC' in *Source data 1*). As a note, while intraday measurement error is small for the TGF-α shedding assay (typically, 1%–2% AP-TGF-α release), interday variability varies widely depending on cell conditions. Since the SD represents interday variability, the basal SD cut-off removes more couplings than the SD cut-off of ligand-induced signal or the significance of individual experiments. As a consequence, some of manually annotated couplings in the Inoue dataset are regarded as noncoupling by the basal SD cut-off criteria, including for $P2RY_2$ and $P2RY_6$ that had no couplings above this cut-off.

## Generating a subset of comparable GPCR-G protein couplings

To enable qualitative comparison of corresponding datapoints in the datasets from the Bouvier (*Avet et al., 2020*) and Inoue (*Inoue et al., 2019*) groups, we used the subset of 70 GPCRs present in both datasets and belonging to the same class, A (removed only two receptors from class B1). The quantitative comparisons focused on a smaller subset of 51 GPCRs tested with the same ligand and excluded noncoupling GPCR-G protein pairs, as they could not be represented by 0 values (e.g. an underrepresentation of couplings in a given datasets and G proteins, see Results). All analyses herein included the 12 G proteins: $G_s$, $G_{i1}$, $G_{i2}$, $G_{oA}$, $G_{oB}$, $G_z$, $G_q$, $G_{11}$, $G_{14}$, $G_{15}$, $G_{12}$, and $G_{13}$. $G_{olf}$ and $G_{i3}$ could not be analyzed, as they had not been tested by the Bouvier group (*Table 1*). The Inoue data for the pairs $G_{i1-2}$, $G_{oA-B}$, and $G_q$ and $G_{11}$, respectively, were generated with identical chimera inserting the Gα C-terminal hexamer into a $G_q$ backbone (*Inoue et al., 2019*). Qualitative analyses compared the

presence or absence of each GPCR-G protein coupling while the quantitative analyses were limited to common G protein couplings, that is, data points in which both the Bouvier and Inoue groups generated a $pEC_{50}$ and $E_{max}$ value.

### Validation of comparability when $E_{max}$ or reference agonist differ

To get an overview of the distribution of data in the two datasets, we determined the $pEC_{50}$ and $E_{max}$ mean, median, min, max, span, and SD values and plotted box and whiskers plots for each G protein (*Source data 1*). This showed that the $E_{max}$ values vary much more across the different G proteins than $pEC_{50}$ values. The $E_{max}$ variation is largest in the Bouvier data wherein $G_{15}$ ranges across three orders of magnitude (16–1067). The Bouvier dataset has low means and spans for $G_s$ (42 and 47) and G12 (71 and 101) indicating a narrow assay signal window (low signal to noise). We find that minimum-maximum normalization (to 100%) of each G protein across receptors gives a more uniform distribution (SDs for Bouvier: 19–31 and Inoue: 22–26, rightmost plot pair in *Source data 1*).

Whereas both studies tested a majority of receptors with their endogenous ligand, surrogate agonists were used for 15 and 4 GPCRs in the data from the Inoue and Bouvier groups, respectively. Although those ligands were selected for their reference character with similar pharmacology to the endogenous ligand, they could introduce differences in a receptor's G protein profile due to ligand-dependent signaling bias. Analysis of GPCR-G protein couplers and noncouplers (tab 'BI' in *Source data 2*) shows that on average 74% and 71% agreeing qualitative couplings (i.e. coupling vs. noncoupling GPCR-G protein pairs) for receptors tested with the same and different agonists, respectively. This is a rather small difference providing confirmation that receptors tested with different ligands can be compared on the qualitative coupling/noncoupling level and be included in the comparison of the different G protein coupling datasets and determination of novel G protein couplings.

### Dataset integration into a unified coupling map – normalization and $\log(E_{max}/EC_{50})$ values

To enable quantitative correlation of the Bouvier and Inoue couplings, we further filtered the 70 common GPCRs to yield 51 common class A GPCRs tested with the same ligand (excluding 29 receptors tested with different ligands and two class B1 GPCRs). To assess the value of normalization, we calculated the average 'similarity' (Bouvier/Inoue ratio) of individual values (tabs ending with '-sim' in *Source data 4*) and the 'linear correlation' ($r^2$ value) of each receptor across all G proteins (reported below as averages of individual couplings and G proteins, respectively) (tabs ending with '-$r^2$' in *Source data 4*). Linear correlation was only done for receptors with at least three common G proteins/families. Minimum-maximum normalized $E_{max}$ values were represented as percentage values while decimal values (0–1) were used for the calculation of $\log(E_{max}/EC_{50})$ values as recommended in *Kenakin, 2017*. For both the Bouvier and Inoue groups, the minimum and maximum represent the signal without (0%) and with (100%) an agonist, respectively, in each experiment replica. The minimum $E_{max}$ value was therefore set 0 while the maximum was set to the highest value for the given G protein (first, use normalization) or receptor (second, tested but not used normalization).

Minimum-maximum normalization of $E_{max}$ measurements increased the average from 0.22 to 0.61 and the linear correlation $r^2$ value from 0.28 to 0.30. In contrast, non-normalized $pEC_{50}$ measurements have the most similar values (0.89 compared to 0.66) and an identical linear correlation (0.37). We combined the minimum-maximum normalized $E_{max}$ and non-normalized $EC_{50}$ values into $\log(E_{max}/EC_{50})$ (*Kenakin, 2017*). The $\log(E_{max}/EC_{50})$ values have an average similarity that is nearly as high (0.86 compared to 0.89) and an average linear correlation that is better (0.41 compared to 0.37) than for $pEC_{50}$, the best individual measure. We did not apply a double normalization, that is, also across G proteins after across receptors, as this worsens the $E_{max}$, $pEC_{50}$, and $\log(E_{max}/EC_{50})$ value similarity (0.61–0.39, 0.89–0.42, and 0.86–0.47, respectively) and linear correlation of each G protein across receptors (0.11–0.05, 0.49–0.20, and 0.46–0.23, respectively).

### Dataset integration into a unified coupling map – G protein family aggregation

Given that G proteins belong to families that are functionally grouped by sharing downstream signaling pathways, we next investigated the best member-to-family aggregation scheme – specifically, whether the most comparable G protein family values are obtained if using the maximum value from any single

subtype or the mean of all subtype members. We found that aggregation using max rather than mean values gives a better similarity for $E_{max}$ (0.69 vs. 0.65), marginally lower similarities for $pEC_{50}$ (0.89 relative 0.90), and identical similarities for $\log(E_{max}/EC_{50})$ (0.88). However, max performs better overall than mean in the comparisons correlating a receptor across G proteins; $E_{max}$ (0.63 and 0.53), $pEC_{50}$ (0.56 and 0.56), and $\log(E_{max}/EC_{50})$ (0.65 and 0.59) or one G protein across receptors; $E_{max}$ (0.29 and 0.20), $pEC_{50}$ (0.71 and 0.74), and $\log(E_{max}/EC_{50})$ (0.69 and 0.71). Notably, the correlation of each receptor across max-aggregated G protein families compared to nonaggregated subtypes is much stronger when considering any of the three pharmacological parameters: $E_{max}$ (0.63 vs. 0.36), $pEC_{50}$ (0.56 vs. 0.30), and $\log(E_{max}/EC_{50})$ (0.65 vs. 0.41). Altogether, this establishes the highest value (max) as the aggregation that yields the most comparable quantitative value for G protein families.

## Tissue expression analysis

Consensus transcript expression levels were extracted for 50 different tissues aggregated into 16 organs or systems in the HPA (*Uhlén et al., 2015*), which also includes data from the GTEx project (*Lonsdale et al., 2013*). The G protein expression was quantified in the form of nTPM, which is a consensus normalized transcript expression value (see https://www.proteinatlas.org/about/assays+annotation#rna). The correlation analysis was performed using a default Pearson correlation analysis using Seaborn's clustermap.

## Statistical Analysis

The aggregated sample size is n = 3 or higher for all analyzed GPCR-G protein couplings. The specific sample size for each such coupling is described in the original articles reporting these data and referenced in the present manuscript. The Bouvier dataset contained some datapoints that were included or excluded based on dedicated analyses (see 'Coupling datasets' above). For all figures, the values for *N*, *P*, and the specific statistical test are included in the figure legend or associated manuscript text. For *Figure 5*, pairwise correlation of $\log(E_{max}/EC_{50})$ values was assessed as a measure of the strength of the linear relationship between two pathways. The pairwise distances between pathways were calculated in python scipy package using the *spatial.distance.pdist* method using the *correlation* metric. Statistical significance was determined by Pearson standard correlation coefficient with a two-tailed p-value as implemented in scipy's *stats.pearsonr* method.

## Acknowledgements

Nevin Lambert is acknowledged for fruitful discussions. We thank Ayumi Inoue (Tohoku University) for reanalyzing the raw data of the TGF-α shedding assay to obtain SD values for the individual GPCR-G protein pairs. Funding: Canada Research Chair in Signal Transduction and Molecular Pharmacology (MB); Canadian Institutes of Health Research grant FDN-148431 (MB); Lundbeck Foundation grants R218-2016-1266 and R313-2019-526 (DEG); Novo Nordisk Foundation grant NNF18OC0031226 (DEG); Japan Society for the Promotion of Science (JSPS) 21H04791, 21H05113 and JPJSBP120213501 (AI); Japan Agency for Medical Research and Development (AMED) JP20gm0010004 and JP20am0101095 (AI); Japan Science and Technology Agency (JST) JPMJFR215T (AI); Takeda Science Foundation (AI); Uehara Memorial Foundation (AI); Daiichi Sankyo Foundation of Life Science (A.I.).

## Additional information

### Competing interests

Claire Normand, Arturo Mancini: was an employees of Domain Therapeutics North America during part or all of this research. Michel Bouvier: is the president of Domain Therapeutics scientific advisory board. The other authors declare that no competing interests exist.

### Funding

| Funder | Grant reference number | Author |
| --- | --- | --- |
| Canadian Institutes of Health Research | FDN-148431 | Michel Bouvier |

| Funder | Grant reference number | Author |
| --- | --- | --- |
| Lundbeckfonden | R218-2016-1266 | David E Gloriam |
| Lundbeckfonden | R313-2019-526 | David E Gloriam |
| Novo Nordisk Fonden | NNF18OC0031226 | David E Gloriam |
| Japan Agency for Medical Research and Development | JP20gm0010004 | Asuka Inoue |
| Takeda Science Foundation | | Asuka Inoue |
| Uehara Memorial Foundation | | Asuka Inoue |
| Japan Society for the Promotion of Science | 21H04791 | Asuka Inoue |
| Japan Science and Technology Agency | JPMJFR215T | Asuka Inoue |
| Daiichi Sankyo Foundation of Life Science | | Asuka Inoue |
| Lundbeckfonden | R278-2018-180 | Alexander S Hauser David E Gloriam |
| Independent Research Fund Denmark | 8021-00173B | David E Gloriam |
| Japan Agency for Medical Research and Development | JP20am0101095 | Asuka Inoue |
| Japan Society for the Promotion of Science | 21H05113 | Asuka Inoue |
| Japan Society for the Promotion of Science | JPJSBP120213501 | Asuka Inoue |

The funders had no role in study design, data collection and interpretation, or the decision to submit the work for publication.

## Author contributions

Alexander S Hauser, Formal analysis, Investigation, Validation, Visualization, Writing – review and editing; Charlotte Avet, Data curation, Formal analysis, Investigation, Writing – review and editing; Claire Normand, Arturo Mancini, Data curation; Asuka Inoue, Formal analysis, Funding acquisition, Writing – review and editing; Michel Bouvier, Conceptualization, Funding acquisition, Supervision, Writing – review and editing; David E Gloriam, Conceptualization, Data curation, Formal analysis, Funding acquisition, Investigation, Methodology, Project administration, Supervision, Validation, Visualization, Writing - original draft, Writing – review and editing

## Author ORCIDs

Alexander S Hauser  http://orcid.org/0000-0003-1098-6419
Asuka Inoue  http://orcid.org/0000-0003-0805-4049
Michel Bouvier  http://orcid.org/0000-0003-1128-0100
David E Gloriam  http://orcid.org/0000-0002-4299-7561

## Decision letter and Author response

Decision letter https://doi.org/10.7554/eLife.74107.sa1
Author response https://doi.org/10.7554/eLife.74107.sa2

## Additional files

### Supplementary files
• Transparent reporting form

• Source data 1. SD cut-off and pEC$_{50}$ and E$_{max}$ distributions. This spreadsheet contains a calculation of a standard deviation (SD) cut-off that gives the best agreement between G protein couplings from the Bouvier and Inoue groups. The cut-off, 1.4 represent the number of standard deviations of a G protein coupling average (at least from triplicates) from the basal signal of the given receptor. The spreadsheet also contains a DataStats tab detailing the distribution of raw and normalized pEC$_{50}$ and E$_{max}$ values.

• Source data 2. Assessment of effect of different ligands. This spreadsheet contains a comparison of the relative average difference obtained for G protein couplings when determined with and without using the same ligand in the Bouvier and Inoue groups.

• Source data 3. Qualitative coupling comparisons. This spreadsheet contains a qualitative comparison (coupling and non-coupling) of GPCR-G protein pairs from the Bouvier and Inoue groups, and from the Guide to Pharmacology database.

• Source data 4. Normalization and aggregation into G protein families. This spreadsheet contains a comparative analysis of which normalization and aggregation into G protein families that gives the best agreement between G protein couplings from the Bouvier and Inoue groups.

• Source data 5. Unified coupling map. This spreadsheet contains a unification of G protein couplings from the Bouvier and Inoue groups, and from the Guide to Pharmacology database into a common coupling map.

## Data availability

All underlying data are available in Spreadsheets S1-5. The obtained common coupling map is available in the online database GproteinDb at https://gproteindb.org/signprot/couplings.

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
