## [Editor Report]

Using data sets are from three distinct sources, the authors present a meta-analysis to arrive at a consensus for G protein-coupled receptor (GPCR) coupling specificity to G proteins, as well as to identify and highlight differences or incongruencies. Compiling these data sets into a unified format will be extremely useful for investigators to understand receptor and effector relationships. The meta-analysis will help to deconvolute the complex physiology and pharmacology underlying hormone or drug actions acting on receptor superfamilies, and a better understanding of receptor-G protein promiscuity will ultimately help in identifying safer therapeutics.

---

## [Decision Letter]

**Decision letter after peer review:**

Thank you for submitting your article "GPCR-G protein selectivity -a unified meta-analysis" for consideration by *eLife*. Your article has been reviewed by 2 peer reviewers, and the evaluation has been overseen by a Reviewing Editor and Richard Aldrich as the Senior Editor. The following individual involved in review of your submission has agreed to reveal their identity: Roger Sunahara (Reviewer #2).

Essential revisions:

The paper should be classified as Tools and Resources.

This work provides a resource of broad interest and utility to the large community of researchers interested in GPCR coupling to G proteins. By combining 2 (published) large datasets of interactions with the widely-used (literature-based) Guide to Pharmacology database it substantially increases both the amount of GPCR/G protein coupling validated by multiple studies and quantifies the interactions better. The data are well analyzed, but there are several concerns that need to be addressed. The principal concerns are that 1) there is no analysis of distinctions within a G subfamily, and 2) there is no demonstration of the utility of the analysis, i.e. demonstrating an unexplored coupling in a physiologically relevant context.

Overall, it is a well-written (but not well-edited) manuscript. There are several typographical errors in the manuscript, perhaps as a result of the choice of font size. It was hard enough for the reviewers to carefully read through the text, and apparently for the authors as well.

1. There are specific properties of some G protein isoforms within each family that need to be considered when making these comparisons. For example, there are vastly different nucleotide exchange properties of Galphao (Gao) from the other Gi isoforms (Gai1, Gai2, Gai3 and Gaz). Moreover, Gao has quite a distinct expression pattern than other isoforms where it is mostly expressed in brain with levels reaching 1%-3% of total membrane protein concentration. These properties are likely related to the capacity of this isoform to exhibit rapid regulation of effectors such as ion channels. Thus, merging the data sets for Gao with other Gai isoforms, particularly when the coupling data are based on recombinant expression in HEK cells, or with Galpha chimeras, should done with caution. One might even consider performing the meta-analysis of Gao separately from the other Gai isoforms.

2. The authors raise an interesting and important caveat in the Bouvier and Inoue screens regarding the EC50 and efficacy differences compared to the GtP screens. It is fine to point out the differences, but possible explanations for the differences, especially when comparing three different screening platforms, needs to be provided.

3. The data are described mostly by broad numbers, such as the number of receptors / coupling in a subset, or by percentages. While this is helpful to understand the data, it is hard to follow the mountain of numbers. The suggestion is to add a section where the authors pick selected examples of particular experimental data and show how their combined database can resolve previously unanswered (or wrongly answered) questions of GPCR/G protein coupling.

4. The paper does not reveal new biological findings, but the authors should be able to do so if they use their new resource for specific examples. For example, while some emphasis is placed on new data on G15, it would be helpful to take the extra step and use this to suggest new biological insights. Looking more deeply into those G proteins that are outliers in their subfamilies, such as Gz, G15, Gt3 would also be useful.

5. The authors cautiously label couplings supported by only one data set as "unsupported". It would seem more helpful to grade couplings by a reliability scale, providing users with a wider set of data. Perhaps only couplings that are directly conflicted by negative data should be labeled as unsupported?

6. Given that this manuscript includes authors from both the Inoue and Bouvier studies, it is somewhat understandable that they are not directly assessing which of the two datasets (in relation to the GtP) might be more accurate. Nevertheless, this assessment should be done and the advantages and disadvantages of the two experimental systems discussed clearly. The authors should use their new resource to assess the weaknesses and strengths of each of their respective experimental methods, especially when there are data in GtP that can be used for comparisons. They briefly mention such a conclusion in line 136-137 or line 142, but this should be done more broadly and more deeply.

7. Lines 218-219 say "The choice of protein systems and biosensors for such future studies should take into account the relative strengths and limitations of the profiling platforms used". Please provide such an assessment here, with the data you provide.

8. The sentence "this shows that G protein family pairs have evolved differing levels of receptor co-coupling or selectivity, which they have achieved by weakening or abolishing the binding to the same GPCRs" in line 209-210 begs a deeper investigation of this using this resource and selected G proteins. This is also potentially confounded by the use of overexpression for many of the examples in the companion paper.

*Reviewer #1 (Recommendations for the authors):*

To emphasize the last weakness, the authors should use their new resource to assess the weaknesses and strengths of each of their respective experimental methods, esp. when there's data in GtP that can be used for comparisons. They briefly mention such a conclusion in line 136-137 or line 142, but this should be done more broadly and more deeply.

Lines 2128-219 say "The choice of protein systems and biosensors for such future studies should take into account the relative strengths and limitations of the profiling platforms used," So please give us such an assessment here, with the data you provide.

Another improvement would be to look more deeply into those G proteins that are outliers in their subfamilies, such as Gz, G15, Gt3. This ties into my comment above, that the manuscript lacks a biological story, one that I believe the authors can reach if they use their new resource for specific examples.

For me, the sentence "this shows that G protein family pairs have evolved differing levels of receptor co-coupling or selectivity, which they have achieved by weakening or abolishing the binding to the same GPCRs" in line 209-210 begs a deeper investigation of this using this resource and selected G proteins.

*Reviewer #2 (Recommendations for the authors):*

This is a timely review/analysis and will be extremely useful for the GPCR field. Overall, it is a well-written (but not well-edited) manuscript that will be greatly appreciated by GPCR biologist. The tables, figures and Gantt charts are useful.

There are specific properties of some G protein isoforms within each family that need to be considered when making these comparison. I draw the authors' attention to the vastly different nucleotide exchange properties of Galphao (Gao) from the other Gi isoforms (Gai1, Gai2, Gai3 and Gaz). Moreover, Gao has quite a distinct expression pattern than other isoforms where it is mostly expressed in brain with levels reaching 1%-3% of total membrane protein concentration. These properties are likely related to the capacity of this isoform to exhibit rapid regulation of effectors such as ion channels. Thus, merging the data sets for Gao with other Gai isoforms, particularly when the coupling data are based on recombinant expression in HEK cells, or with Galpha chimeras, should done with caution. One might even consider performing the meta-analysis of Gao separately from the other Gai isoforms.

The authors raise an interesting and important caveat in the Bouvier and Inoue screens regarding the EC50 and efficacy differences compared to the GtP screens. One issue is to point out the differences but at some point an explanation for the differences need to be addressed, especially when comparing three different screening platforms.

---

## [Author Response]

Essential revisions:The paper should be classified as Tools and Resources.

The meta-analysis unifying datasets into a common coupling map fits the Tools and Resource format very well. The data normalization protocol (leading to log(Emax/EC50) as a comparable parameter), explanation and overview of the couplings are all tools/resources that are unique to this paper, whereas the resulting coupling map has recently been published as an interactive “G protein couplings” browser in the GproteinDb database (1). Furthermore, we have edited the manuscript to reduce its technical nature.

Beyond the Tools and Resources component, the coupling map allowed us to advance GPCR-G protein selectivity with new or updated findings that instead would fit with the publication format “Research Advances” (https://reviewer.elifesciences.org/author-guide/types). This part has now been expanded with analyses of G protein expression profiles and co-expressions. We have clarified the dual nature in the revised manuscript by splitting Results into two subsections named “Tools and Resources – Common coupling map unifying GPCR-G protein data” and “Research Advances – Insights on GPCR-G protein selectivity”. This has also been clarified in the title, which has been updated to “Common coupling map advances GPCR-G protein selectivity”.

Given this dual content, we would please like to ask for an updated opinion on the most suitable publication format based on the revised manuscript. Would it be most suitable as a Tools and Resource, Research Advances or Research Article (to cover both aspects)?

This work provides a resource of broad interest and utility to the large community of researchers interested in GPCR coupling to G proteins. By combining 2 (published) large datasets of interactions with the widely-used (literature-based) Guide to Pharmacology database it substantially increases both the amount of GPCR/G protein coupling validated by multiple studies and quantifies the interactions better. The data are well analyzed, but there are several concerns that need to be addressed. The principal concerns are that 1) there is no analysis of distinctions within a G subfamily, and 2) there is no demonstration of the utility of the analysis, i.e. demonstrating an unexplored coupling in a physiologically relevant context.Overall, it is a well-written (but not well-edited) manuscript. There are several typographical errors in the manuscript, perhaps as a result of the choice of font size. It was hard enough for the reviewers to carefully read through the text, and apparently for the authors as well.

We have increased and differentiated the font size of headings, adjusted the page margin where incorrect and updated the text in several places to increase legibility, including those pointed out by Reviewer 1. We have also rewritten several sections of Results and paragraphs in the Discussion to increase clarity.

Reviewer #1 (Recommendations for the authors):To emphasize the last weakness, the authors should use their new resource to assess the weaknesses and strengths of each of their respective experimental methods, esp. when there's data in GtP that can be used for comparisons. They briefly mention such a conclusion in line 136-137 or line 142, but this should be done more broadly and more deeply.Lines 2128-219 say "The choice of protein systems and biosensors for such future studies should take into account the relative strengths and limitations of the profiling platforms used," So please give us such an assessment here, with the data you provide.

We believe that the three-way intersection of couplings is the most informative and therefore preferred over individual comparison of each of the Inoue and Bouvier datasets to GtP. GtP is unfortunately not suitable as a stand-alone resource – neither to contradict nor support couplings (on the G protein subtype level). This is because GtP is incomplete (especially for G_12/13_) and does not provide any information on the level of G protein subtypes, only families. The three-way interactions will always use GtP but adds a second dataset on top of this when validating a third dataset. Our manuscript already included a three-way intersection of datasets, allowing readers to conclude which dataset might be more accurate (then Figure 3 and Spreadsheet 3) on a per-G protein basis.

In the revised manuscript, we have rewritten this section, which now has the heading “Bouvier’s and Inoue’s biosensors appear more sensitive for G_15_ and, G_s_ and G_12_, respectively. We have also made a completely new figure, Figure 7, which more clearly illustrates for which G proteins that Bouvier and Inoue may have overrepresented or underrepresented couplings. This section specifically investigates the question of whether differential sensitivity can explain “unique” couplings. However, such unique couplings can either be due to overrepresentation or instead be true positives that are missing in GtP because of incompleteness and in the other biosensor due to lower sensitivity. Unfortunately, we will not be able to distinguish these possibilities until the research community has gained additional datasets from independent biosensors with as high sensitivity.

Whereas our study compares datasets rather than experimental systems, we have added a paragraph in the Discussion describing which aspects should be considered when choosing a biosensor. There, we reference a review from last year dedicated to biosensors and describing their pros and cons (3), and the accompanying paper by Bouvier et al. (4), comparing several aspects of the experimental system used by Inoue et al. It is also important to note that the most advantageous biosensor may be one of the two for which data is analyzed in our paper. For many studies, researchers may instead be better off with another biosensor, for example those from Lambert/Mamyrbekov (5), Roth (2) (Gαβγ sensors first described in (6-11)) or Inoue (unpublished dissociation assays using wt G proteins fused with LgBit and HiBit). These are all referenced in the Discussion.

Additionally, the conclusions in lines 136-137, 142 and 218-219 have now been better described as part of a rewriting of the respective section of Results, “Bouvier’s and Inoue’s biosensors appear more sensitive for G_15_ and, G_s_ and G_12_, respectively”.

Another improvement would be to look more deeply into those G proteins that are outliers in their subfamilies, such as Gz, G15, Gt3. This ties into my comment above, that the manuscript lacks a biological story, one that I believe the authors can reach if they use their new resource for specific examples.

We have now added an additional biological story describing the expression profiles and co-expression of G proteins. This section describes which G proteins are outliers in terms of expression, including G_z_ and G_t3_, which had not been included previously, as there is no coupling data for this G protein. Furthermore, we have increased and clarified the usage of particular examples of G proteins (and GPCRs) within the different sections of the Results subsection “Research Advances – Insights on GPCR-G protein selectivity”.

With the public data currently available, we feel that this is the best balance when avoiding going to far on a limited data basis versus covering the relevant, supported biological stories of G proteins. It would be advisable to await further independent datasets from high-sensitivity biosensors before looking more deeply are outlier couplings. This is because GtP does not provide subtype data, and the outliers are typically where the Bouvier and Inoue datasets differ largely. Therefore, a more detailed analysis may risk not being reproducible until additional datasets become available.

For me, the sentence "this shows that G protein family pairs have evolved differing levels of receptor co-coupling or selectivity, which they have achieved by weakening or abolishing the binding to the same GPCRs" in line 209-210 begs a deeper investigation of this using this resource and selected G proteins.

This section has now been rewritten to better investigate and distinguish the contributions of binding/non-binding and weak/strong binding, respectively as mechanisms of GPCR-G protein selectivity.

Reviewer #2 (Recommendations for the authors):This is a timely review/analysis and will be extremely useful for the GPCR field. Overall, it is a well-written (but not well-edited) manuscript that will be greatly appreciated by GPCR biologist. The tables, figures and Gantt charts are useful.There are specific properties of some G protein isoforms within each family that need to be considered when making these comparison. I draw the authors' attention to the vastly different nucleotide exchange properties of Galphao (Gao) from the other Gi isoforms (Gai1, Gai2, Gai3 and Gaz). Moreover, Gao has quite a distinct expression pattern than other isoforms where it is mostly expressed in brain with levels reaching 1%-3% of total membrane protein concentration. These properties are likely related to the capacity of this isoform to exhibit rapid regulation of effectors such as ion channels. Thus, merging the data sets for Gao with other Gai isoforms, particularly when the coupling data are based on recombinant expression in HEK cells, or with Galpha chimeras, should done with caution. One might even consider performing the meta-analysis of Gao separately from the other Gai isoforms.

If not mistaken, Gz also has a slower GTPase activity than the other Gi/o family members?

In the revised manuscript, we have added a section “Differential tissue expression gives G proteins in the same family large spatial selectivity” that brings up the point of differential expression. This is revisited in a new paragraph of the Discussion, which includes the text:

“The best way to address this would be for future studies to profile the couplings of natively expressed GPCRs and G proteins across different cell types. Meanwhile, a strategy could be to analyze GPCR-G protein selectivity for a subset of G proteins that are selected based on the highest expression in the tissue of interest. Such separated meta-analyses of G proteins may be facilitated by the filtering options in the coupling maps in Spreadsheet S5 and online in the G protein couplings browser in GproteinDb (1).“

The same paragraph also mentions the differential nucleotide exchange of Gao and Gaz relative to other members of the Gi/o family. We did not perform a meta-analysis of Gao or Gao separately from the other Gai isoforms, for consistency, length limitations and because the signaling may be inseparable in some tissues. However, readers that wish to do so could do that by using the G protein couplings browser (https://gproteindb.org/signprot/couplings) in GproteinDb (1) or by modifying Spreadsheet S3 in MS Excel.

The authors raise an interesting and important caveat in the Bouvier and Inoue screens regarding the EC50 and efficacy differences compared to the GtP screens. One issue is to point out the differences but at some point an explanation for the differences need to be addressed, especially when comparing three different screening platforms.

As described in the response to Reviewer 1, the differences of screening platforms in terms of their datasets and sensitivity are described in the section “Bouvier’s and Inoue’s biosensors appear more sensitive for G_15_ and, G_s_ and G_12_, respectively”. That response also mentions that aspects to consider for the choice of experimental system are now listed in a new paragraph of Discussion. This paragraph references comparisons in the accompanying paper (4) and a recent review describing the pros and cons for the variety of biosensors available (3).

Furthermore, the difference span not only the screening platforms but also the data annotated from literature in GtP. This is noted in the abovementioned Discussion paragraph in the sentence:

“Of note, most studies annotated in GtP have used downstream measurements, which are amplified, and may therefore differ from those obtained by biosensors that measure G protein activation (further discussed in (4)).”